# Colorimetric Determination of Glucose in Sweat Using an Alginate-Based Biosystem

**DOI:** 10.3390/polym15051218

**Published:** 2023-02-28

**Authors:** Sandra Garcia-Rey, Eva Gil-Hernandez, Lourdes Basabe-Desmonts, Fernando Benito-Lopez

**Affiliations:** 1Microfluidics Cluster UPV/EHU, Analytical Microsystems & Materials for Lab-on-a-Chip (AMMa-LOAC) Group, Analytical Chemistry Department, University of the Basque Country UPV/EHU, 48940 Leioa, Spain; 2Microfluidics Cluster UPV/EHU, BIOMICs Microfluidics Group, Lascaray Research Center, University of the Basque Country UPV/EHU, 01006 Vitoria-Gasteiz, Spain; 3Basque Foundation of Science, IKERBASQUE, Calle María Díaz de Haro 3, 48013 Bilbao, Spain

**Keywords:** sweat glucose, colorimetric analysis, RGB, microfluidic

## Abstract

Glucose is an analyte of great importance, both in the clinical and sports fields. Since blood is the gold standard biofluid used for the analytical determination of glucose, there is high interest in finding alternative non-invasive biofluids, such as sweat, for its determination. In this research, we present an alginate-based bead-like biosystem integrated with an enzymatic assay for the determination of glucose in sweat. The system was calibrated and verified in artificial sweat, and a linear calibration range was obtained for glucose of 10–1000 µM. The colorimetric determination was investigated, and the analysis was carried out both in the black and white and in the Red:Green:Blue color code. A limit of detection and quantification of 3.8 µM and 12.7 µM, respectively, were obtained for glucose determination. The biosystem was also applied with real sweat, using a prototype of a microfluidic device platform as a proof of concept. This research demonstrated the potential of alginate hydrogels as scaffolds for the fabrication of biosystems and their possible integration in microfluidic devices. These results are intended to bring awareness of sweat as a complementary tool for standard analytical diagnosis.

## 1. Introduction

Glucose is the major energy source for cells, and its levels are regulated in blood by homeostasis [1]. Therefore, glucose monitoring is of high importance in order to detect a great variety of disease conditions, such as hypoglycemia and diabetes [2]. However, blood is still the gold standard biofluid for clinical diagnosis, including the determination of sugar-related alterations and diseases. Due to the invasiveness involved, research of less invasive biofluids, such as tears, saliva, sweat and urine, has been recently prioritized [3,4]. 

Among these biofluids, sweat has taken the lead due to its high potential. Since sweat is generated by every individual and is easily accessible, sweat arises as a potential alternative diagnostic tool for blood analysis for glucose determinations. Despite its clinical applicability, glucose sweat determination also provides a useful tool in the sports field, since it can be used to prevent health issues that may arise during exercise, as well as to improve the performance of athletes. Glucose levels in sweat are established to be between 10 µM–1000 µM, with the physiological range being 60 µM–110 µM [5,6].

Different materials have been recently developed for glucose sensing, which comprise a varying range of sensing platforms that go from materials with enzymatic activity to nanofibers. In fact, Yu et al. immobilized a fluorescent carbon quantum dots film with glucose oxidase and a cellulose acetate complex into the tip of an optical fiber, and they were able to detect glucose of 0.01–0.1 µM, obtaining a limit of detection of 0.026 µM [7]. Another different approach for glucose sensing developed by Wiorek et al. was based on the fabrication of an epidermal patch for the determination of glucose in sweat, and they were able to detect glucose in a linear calibration range of 10–200 µM [8]. The same year, Cao et al. developed a paper-based microfluidic approach for the determination of glucose, which was integrated with an electrochemical biosensor based on reduced graphene oxide-tetraethylene pentaamine [9]. Following this approach, they were able to detect glucose in a linear range of 0.1–25 mM, with a detection limit of 25 µM. 

Although electrochemical detection offers a higher sensitivity [10], colorimetric detection offers a simple, easy-to-read and cost-affordable approach for the determination of biomarkers, as it allows the performance of qualitative determinations by naked eye or the integration of optical detectors for quantitative measurement [11]. However, the real applicability of sweat glucose determination towards the sports field relies on the integration of these biosystems into wearable microfluidic platforms that allow the performance of the devices to be carried out in field. In this regard, the coupling of the biosensors with smartphones has become the main approach due to their computing power, portability, cost-effectiveness, ease of operation, large memory, adequate battery, powerful computation capability, large data storage, portable power supply and convenient user interface [12].

In this regard, several microfluidic platforms have been developed for the measurement of glucose with a colorimetric approach. For example, He et al. fabricated a thermo-responsive, textile/paper-based, microfluidic analysis system by combining biocompatible polyurethane, cotton fabric and a paper-based colorimetric sensor, which was coupled with a smartphone, for the quantitative determination of sweat glucose, obtaining a limit of detection of 13.49 µM [13]. Also coupled to a smartphone, Xiao et al. fabricated a PDMS device for the colorimetric detection of sweat glucose [14]. This platform allowed 5 parallel determinations in a single measurement, measuring sweat glucose in a linear range of 0.1–0.35 µM, with a limit of detection of 30 µM. Another common approach for sweat glucose determination is the development of multiplexed analysis, in which glucose is detected together with other relevant sweat biomarkers. In this regard, Choi et al. fabricated a multilayer microfluidic device coupled with a smartphone for the colorimetric determination of chloride, glucose and lactate across physiologically relevant ranges [15]. Moreover, this platform integrated capillary bursting valves in the microfluid layer for the sequential filling of the reservoirs, thus allowing sequential sampling of sweat. 

Recently, alginate hydrogels have proven to be appropriate scaffolds for the detection of metabolites in sweat, with a high potential to be integrated into wearable microfluidic platforms since they are non-toxic, biocompatible, biodegradable and affordable materials. In this regard, in previous research, we developed a proof-of-concept alginate bead biosystem for the detection of lactate in sweat, obtaining a linear calibration range for lactate of 10–100 µM [16]. Lactate was detected at 13 min, and the biosystem was validated with real sweat samples. Moreover, we have also demonstrated the stability of an alginate-based biosystem integrated with an enzymatic reaction for the detection of glucose and lactate, in which the sensing capabilities of the material were maintained for at least 10 days [17]. Other approaches for glucose sensing involving alginate hydrogels include enzymatic immobilization in alginate-based microfibers [18], luminescent cupper nanoclusters in alginate [19] or electrodepositable alginate membranes for the amperometric determination of glucose [20]. 

Based on these results, we have developed a proof-of-concept alginate-based biosystem integrated with an enzymatic assay for the detection of glucose in sweat through a colorimetric approach. For the calibration of the biosystem, black and white (B/W) analysis and analysis in the red, green and blue color code (RGB) were investigated. Moreover, to demonstrate the potential of this biomaterial, a proof-of-concept poly(methyl methacrylate) (PMMA) device was fabricated for the verification of the alginate biosystem. This research aims at improving the available tools for the determination of biomarkers in sweat, reinforcing the relevance of this biofluid as a potential alternative for diagnostic analysis. 

## 2. Materials and Methods

### 2.1. Artificial Sweat

Artificial sweat was prepared by dissolving NaCl 60 mM (≥99.5%, Sigma-Aldrich, Madrid, Spain) and urea 60 mM (99%, Sigma-Aldrich, Madrid, Spain) in distilled water [16]. Solutions of artificial sweat with *D*-(+)-glucose (≥99.5%, Sigma-Aldrich, Madrid, Spain) 10, 20, 40, 50, 60, 80, 100, 150, 250, 500, 600, 750 and 1000 µM were made, and the final pH was adjusted to 6.5. The solutions were stored between 2–8 °C until use.

### 2.2. Fabrication of the Biosystem

The alginate beads were fabricated by mixing 5 µL of glucose oxidase (GOx) 0.8 mg mL^−1^ (*Aspergillus niger*, Sigma-Aldrich, Madrid, Spain), 5 µL of horseradish peroxidase (HRP) 0.04 mg mL^−1^ (Sigma Aldrich, Madrid, Spain) and 1.5 µL of tetramethylbenzidine (TMB) (Sigma Aldrich, Madrid, Spain) dissolved in dimethyl sulfoxide (DMSO, Sigma-Aldrich, Madrid, Spain) with 30 µL of sodium alginate (Sigma-Aldrich, Madrid, Spain) 1.5% (*w/v*) in distilled water. TMB was prepared by dissolving 10.7 mg in 1 mL of DMSO. 

For the formation of the beads, 25 µL of the mix were taken and dropped into a 400 mM CaCl_2_ solution (≥93.0% Sigma-Aldrich, Madrid, Spain). The beads were immediately formed. Afterwards, the newly formed beads were washed with distilled water for 3 min before being wiped.

### 2.3. Calibration and Verification of the Biosystem for Glucose Sensing in Plate

For the calibration of the biosystem, a 96-well white plate (Non-Treated Surface, Thermo Fisher Scientific, Madrid, Spain) was used. After placing one bead per well, 150 µL of the glucose solutions in artificial sweat were added to the alginate beads. Four different beads were measured for each glucose concentration that was tested (n = 4).

The experiments were recorded with a 64 MP camera (Sony IMX682 1/1.73”, f/1.89, PDAF, Tokyo, Japan) in a white chamber under the same light conditions. Images at 1, 2, 3, 4, 5, 6, 7, 8, 9, 10, 13, 16, 19, 25, 30, 35 and 40 min were subtracted from the video and were analyzed afterwards with the image processing program ImageJ (1.53t 24 August 2022, NIH), both in the grey scale and in the RGB color code. For each specific time, the same image was used for the four colorimetric analyses and was modified accordingly with ImageJ. For the grayscale analysis, the image was turned into an 8-bit image, and the black-and-white value (B/W value) of each bead was measured, where 0 stands for black and 255 for white. To make the interpretation easier, the measured values were subtracted from 255, so that the more color, the higher the value. For the RGB analysis, the red, green and blue values of each bead were measured and recalculated as before. It needs to be noted that RGB (0,0,0) stands for white, while RGB (255,255,255) stands for black. For the B/W, R, G and B analysis, the whole bead was measured, without including the surrounding media. 

For the verification of the biosystem, which was also done in plate, unknown glucose concentrations were spiked in artificial sweat. Then, 150 µL of those solutions were added to individual alginate beads. After the colorimetric assay was performed and the colorimetric signal was measured, the obtained values were used to determine the glucose concentration of those solutions by using the calibration curves previously obtained. Three individual beads were fabricated and measured for each glucose concentration (n = 3). Moreover, the accuracy and the precision of the obtained values were discussed, attending to the standard deviation (SD) of the measured black and white (B/W) values, compared to the real glucose concentration. The entire process is summarized in Figure 1.

### 2.4. Alginate Biosystem for Glucose Sensing in the Device with Real Sweat

The biosystem was applied with real sweat in a microfluidic platform, which was fabricated by a lamination process. The different layers of the device were, from bottom to top, 229 µm-thickness PSA 8939 (ARcare^®^ 8939, Adhesive Research, Limerick, Ireland), 0.175 mm-thickness PMMA (ME30-SH-000116, clear, Goodfellow, Microplanet Lab., Barcelona, Spain), 244 µm-thickness PSA 90880 (ARseal^™^ 90880, Adhesive Research, Limerick, Ireland), 4.0 mm-thickness PMMA (ME303040, clear, Goodfellow, Microplanet Lab., Barcelona, Spain) and PSA 90880. PSA layers were designed with CorelDRAW Graphics Suite X7 and cut using a Graphtec cutting Plotter CE6000-40 (CPS Cutter Printer Systems, Barcelona, Spain). PMMA layers were designed with CorelDRAW Graphics Suite X7 and cut with a CO_2_ Laser System (VLS2.30 Desktop Universal Laser System, VERSA Laser, Vienna, Austria). 

Before closing the microfluidic platforms with the top PSA layer, alginate beads were placed in the reservoirs of individual devices. The sweat sample was loaded into a syringe, and a positive flow of 10 µL min^−1^ was applied, using a syringe pump (Harvard Apparatus Pump 11 Elite, Madrid, Spain) to drive the flow into the device. The syringe was connected to the device through transparent silicone capillary tubing (ID 1 mm, OD 3 mm, Fisher Scientific, Spain) and PMMA female Luer-Loks (ChipShop, Jena, Germany), which were placed on a circular-shaped PSA 90880 piece surrounding the inlet. Three different devices were used for the verification of the biosystem in the device (n = 3). After the colorimetric analysis of the images, the calibration curve was used for the determination of the glucose concentrations of the sample. This result was compared with a commercially available glucometer (Freestyle Freedom Lite, Abbot, Madrid, Spain), using Freestyle Lite test strips for glucose determination.

## 3. Results and Discussion

### 3.1. Characterization and Working Principle of Alginate Beads

Glucose detection was carried out by integrating an enzymatic assay consisting of GOx, HRP and TMB into an alginate scaffold. A total of 17 µL of the enzymatic mix were taken for the fabrication of the alginate biosystem. The resulting beads were spherical, transparent and had a diameter of 2.4 ± 0.1 mm (n = 10). 

The working principle of this biosystem for glucose detection goes in line with our previous research, where we were able to detect lactate in sweat at 13 min using an alginate-based detection biosystem [16]. Likewise, when glucose is added to the alginate beads, it diffuses inside the biosystem due to the porosity of the alginate hydrogel. This phenomenon can be better understood taking into consideration the mesh size of the alginate biosystem, which has been previously reported to be 6–14 nm [21]. Once inside, the GOx catalyzes the oxidation of the glucose, yielding gluconic acid and H_2_O_2_. Then, this H_2_O_2_ is used by the HRP as the electron donor for the oxidation of the TMB, which is oxidized to a diimine, and it can undergo either a one- or two-electron oxidation. The first colored product, the resulting product of the one-electron oxidation, consists of a charge-transfer complex formed by the diamine, namely, its oxidized diimine product and the radical cation, both species existing in equilibrium. This product absorbs light at 370 nm and 652 nm, respectively, which provides the material with its characteristic blue color. Moreover, it can undergo a further two-electron oxidation, yielding a diamine, which absorbs visible light at 450 nm, generating an orange/yellow-colored product [22]. 

### 3.2. Colorimetric Analysis of the Biosystem

For the calibration of the biosystem, alginate beads integrated with the enzymatic mix were placed in individual wells of a 96-well white plate, and 150 µL of solution with glucose of 10, 20, 40, 60, 80, 100, 250, 500, 750 or 1000 µM in artificial sweat were added. A blank solution was used as a control, which consisted of artificial sweat without glucose. Afterwards, the biosystem was analyzed for 40 min, and the B/W and RGB values of the beads were measured using image analysis software (see Section 2.3). 

Figure 2A shows the measured B/W values of the beads for glucose of 10–1000 µM for 40 min. Higher glucose concentrations yielded higher measured values, which was indicative of more TMB that was being oxidized due to a higher amount of glucose inside the beads; thus, the beads showed a higher-intensity blue color. Moreover, as the assay continued operating, more glucose continued entering the alginate beads, which led to a higher-intensity blue color of the beads for each glucose concentration, until the saturation of the system. Figure 2B shows real images of an alginate bead throughout 40 min, after 80 µM of glucose solution was added to artificial sweat. As can be observed, the color intensity started from the outer surface of the bead towards its center, which was due to the added sweat solution not covering the whole bead. The yellowish color surrounding the beads was an indication that part of the reaction was taking place outside the beads. Due to the porosity of the alginate scaffold, while glucose entered the bead, part of the components of the enzymatic mix diffused outside the biosystem, yielding an amount of glucose to be oxidized in the surrounding medium of the bead. However, this did not affect the colorimetric assay since only the spherical shape of the bead was considered to carry out the colorimetric analysis of the biosystem. 

In order to evaluate the best colorimetric sensing approach for our biosystem, we also investigated the RGB values of the beads for glucose of 10–1000 µM over 40 min (see Section 2.4). As expected, the red analysis (Figure 3A) offered the least accurate results since the colorimetric change of the beads was not towards the red. In fact, the different glucose concentrations that were analyzed with this approach did not show an appropriate resolution among them, which led to similar red values for different glucose concentrations. Surprisingly, however, the green analysis (Figure 3B) showed the best results, while the blue analysis (Figure 3C) was also discarded. It can be assumed that, since it was the blue color intensity of the beads that could be observed by the naked eye to increase over time, the blue analysis would lead to a better colorimetric analysis. However, although the glucose concentrations were better resolved than with the red analysis, large error bars were obtained when the blue values were measured. A possible explanation was that the development of the reaction was not only yielding the first oxidation product of the TMB (blue), but also reaching the second oxidation state (orange/yellow) within the bead. This statement was demonstrated when the green analysis of the alginate biosystem was carried out. In fact, it was this approach that offered the best resolution among the different glucose concentrations, as well as the lowest error bars. 

This could be explained due to the two oxidations that the TMB can undergo, since the first oxidation yields a blue-colored product, while the second oxidation yields a yellow-colored product, with both oxidation states being reversible [22]. If the TMB would have undergone just the first oxidation, thus yielding the blue colored product, the blue analysis would have been a more efficient analysis approach. Therefore, although we could only appreciate the blue color of the beads with the naked eye, the colorimetric analysis of the green values demonstrated that, in fact, a small fraction of the TMB was in the two-electron oxidation state. Consequently, the mix coloration, i.e., the blue-colored product of the TMB with a smaller fraction of the yellow-colored product, led to a more accurate analysis in the green channel. 

### 3.3. Calibration and Verification of the Alginate Biosystem for Glucose Sensing in Plate

Since the B/W and G analysis offered the best resolution and the lowest errors between the tested glucose concentrations, we decided to proceed with these two approaches to evaluate the most appropriate one for the calibration of the biosystem. For both analyses, the detection time was stablished at 13 min since this was when the glucose concentrations can be discriminated among them for the first time. In fact, the calibration distribution followed the same trend for both detection approaches, as can be observed in Figure 4.

Figure 4A shows the calibration curve for the B/W analysis of the beads for glucose of 10–1000 µM at 13 min, which is described by Equation (1). This data strengthens our previously reported results for lactate sensing using an alginate bead biosensor [16], reinforcing the potential of this biosensing material for analytical determinations. Since the glucose solutions tested covered a wide range of concentrations, the calibration curve was divided into two different calibrations curves for a better interpretation of the data, one for the lowest glucose concentrations and another one for the highest glucose concentrations. Following this approach, we obtained a linear calibration range for glucose of 10–100 µM at 13 min (Figure 4B), while a logarithmic calibration was obtained for glucose of 100–1000 µM (Figure 4C) due to the saturation of the system. Similarly, Figure 4D shows the calibration curve for the G analysis of the beads for glucose of 10–1000 µM at 13 min, which is described by Equation (2). The calibration followed the same trend as for the B/W analysis, which was able to detect glucose of 10–100 µM in a linear range at 13 min, while glucose of 100–1000 µM could be detected in a logarithmic approach.
(1)$$y=376.0+\frac{13.4-376.0}{1.0+{\left(\frac{x}{502.4}\right)}^{0.4}}$$(2)$$y=270.2+\frac{60.7-270.2}{1.0+{\left(\frac{x}{257.3}\right)}^{0.9}}$$

The limit of detection (LOD) and limit of quantification (LOQ) for the biosystem were also calculated using the equations LOD=3 SD/m and LOQ=10 SD/m, respectively, where SD is the standard deviation of the blank and m, the slope. An LOD and LOQ of 3.8 µM and 12.7 µM, respectively, were obtained for the B/W analysis and of 7.1 µM and 23.7 µM, respectively, for the G analysis. Therefore, although both the B/W and G colorimetric analyses offered a good approach for glucose determination, the B/W analysis provided a lower LOD and, thus, a better resolution of the biosystem. 

In fact, similar LODs were achieved with this biosystem when compared to electrochemical approaches. For example, Oh et al. [23] fabricated a stretchable and skin-attachable electrochemical sensor for detecting glucose and pH in sweat, obtaining a limit of detection of 1.3 µM for glucose. Later, Shu et al. [24] developed a noninvasive, fiber-material-based, wearable electrochemical sensor to continuously monitor glucose and obtained a limit of detection of 3.28 µM for glucose. Therefore, the alginate biosystem developed here shows a good operational approach and sensitivity, as it is able to compete with previously reported results to be applied for glucose monitoring in sweat. 

Verification of the biosystem for the B/W analysis was also carried out. Glucose solutions of 50, 150 and 600 µM in artificial sweat were prepared, and 150 µL were added to the beads. After 13 min, their B/W value was measured, and their real concentrations were calculated using the calibration curves described in Figure 4B,C, obtaining a concentration of 41 ± 4, 135 ± 26 and 584 ± 34 µM, respectively (n = 3). Comparing the measured concentrations with the real ones, accuracy and precision of at least 83% and 81%, respectively, were obtained for this biosystem. Considering the deviation of the measured values, more precise determinations could be achieved when measuring higher glucose levels compared to the precision of the measurement for glucose of 600, 150 and 50 µM, with a precision of 94%, 81% and 90%, respectively. Similarly, a better accuracy (97%) was achieved for the determination of higher glucose concentrations (see Section 2.3). These results demonstrated the potential of this biosensing material to be used as a new tool for sweat sensing, enhancing the possibility of this non-invasive biofluid to become a complement to standard analytical determinations.

### 3.4. Integration and Application of the Biosystem in a Microfluidic Platform with Real Sweat

Ideally, to get the most out of sweat analysis, detection biosystems should be integrated into microfluidic wearable platforms, which allow sweat biomarker determination to be carried out in the field, at the time and place needed [25]. Therefore, to further demonstrate the potential of this approach in the sports field, the applicability of the alginate biosystem for glucose sensing was carried out in a microfluidic platform prototype with a real sweat sample (Figure 5). It needs to be pointed out that, at this state of investigation, the microfluidic platform was not developed to be implemented as a wearable device, but rather as a microfluidic platform for sweat analysis with the integrated biosystem and as a possible laboratory instrument. It was fabricated by lamination (see Section 2.4) and had a total of five layers of PMMA and PSA, as shown in Figure 5A. The bottom PSA layer, which provided the device with a white color, was added to give a better contrast and thus, to enhance the colorimetric analysis. The device was 25 × 14 × 4.5 mm and consisted of a 2 mm diameter input reservoir, a 400 µm microfluidic channel, a 4 mm diameter sample reservoir and five 100 µm air channels (Figure 5B). These were added to the design in order to avoid the formation of air bubbles inside the microfluidic platform, which allowed the air to flow out of the device while sweat flowed towards the sample reservoir. Figure 5C shows the dimensions of the microfluidic PMMA layer. 

The alginate beads integrated with the enzymatic mix were placed in individual microfluidic platforms prior to closing the device with the top PSA layer. To demonstrate the applicability of the biosystem, 50 µL of sweat were collected from the forehead of a healthy individual after 45 min of indoor cycling. The sample was diluted two times in order to have enough volume for the accomplishment of the assays (enough volume to cover the bead). Afterwards, the sweat was flowed through the microfluidic devices at 5 µL min^−1^ to simulate the low sweat rate of the eccrine glands [26].

Since it was unknown whether the glucose concentration of the sample was in the lower or higher concentration range of the calibration curve, the calibration curve that covered the glucose range of 10–1000 µM was chosen. Equation (1) was used for the calculation of the real glucose concentration at 13 min, obtaining a value of 27 ± 5 µM (n = 3). Figure 5D shows real images of the detection of glucose inside the device at 0, 13 and 20 min, in which the increasing color intensity of the beads is appreciable as an increase of the opacity of the bead. The truly blue color of the bead is not appreciable by the naked eye in this experiment, mainly due to the low concentration of glucose in the sweat sample. Although glucose concentration has great variability among individuals [6], the glucose value measured in this research goes in line with previously reported results. For example, Choi et al. [15] measured glucose in sweat from devices mounted on the forearm across 9 human trials, obtaining glucose concentrations between 4 and 40.4 μM. 

The measured concentration was compared with a commercially available glucometer (see Section 2.4). Nevertheless, the glucometer range was 1100–27,800 µM, which was much higher than the tested glucose concentrations of the real sweat sample; thus, it did not yield a valid result. However, the fact that it was possible to obtain a measurable result with the biosystem, using real sweat in a microfluidic device prototype, demonstrated the applicability of this alginate biosystem as a proof-of-concept for the determination of glucose in real field scenarios. The deviation of the measured concentration could be due to the lower sweat volume that entered the well of the device compared to the 150 µL that were added during the plate study, permitting a lower amount of glucose to be available to enter the bead. Therefore, further research needs to be done before integrating this biosystem in a wearable microfluidic platform.

## 4. Conclusions

We have developed an alginate-based biosystem integrated with an enzymatic assay for the determination of glucose in sweat. Sweat was detected at 13 min in a linear calibration range for glucose of 10–1000 µM, reinforcing our previous research, in which lactate in sweat was calibrated and detected in a similar alginate biosystem [16]. Moreover, the sensing capabilities of this material could be improved by integrating TiO_2_ nanotubes, which has been reported to provide the alginate with a higher hydrophilic character, thus allowing faster detection times [17]. Therefore, this demonstrates the potential of alginate hydrogels as scaffold materials for the fabrication of biosystems. 

The colorimetric analysis of the biosystem was also investigated in both the B/W and the RGB color codes. Surprisingly, although the color change was towards the blue, observable by the naked eye, the measured B values offered poor resolution and showed high errors. Nevertheless, an appropriate calibration of the biosystem was achieved with the B/W and the G values. 

Although further research needs to be done in this field, since this technology is not mature enough to reach the potential end users, the alginate hydrogel biosystem developed in this research has a great potential to be integrated in wearable platforms for sports applications, as demonstrated when integrating the biosystem in a microfluidic device platform prototype. The final device should be flexible to allow correct placement on the skin, biocompatible and user-friendly. Other challenges, such as sample collection and integration of a wireless communication system, would need to be addressed [27,28]. This research is widening the available tools for the fabrication of wearable technology towards the detection of biomarkers in sweat. 

## Figures and Tables

**Figure 1 polymers-15-01218-f001:**
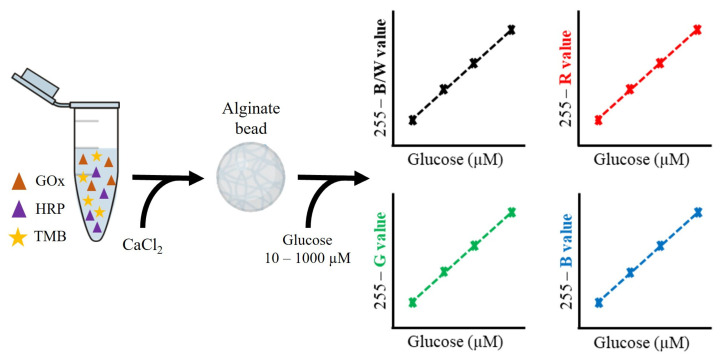
Workflow of the colorimetric analysis approach for the detection of glucose in alginate beads. First, the biosystem is fabricated by mixing glucose oxidase (GOx), horseradish peroxidase (HRP) and tetramethylbenzidine (TMB) with alginate. Then, 25 µL of the mix are dropped into a 400 mM CaCl_2_ solution and the beads are formed. After adding glucose 10–1000 µM, the B/W and the RGB values of the beads are measured for the calibration and verification of the biosystem.

**Figure 2 polymers-15-01218-f002:**
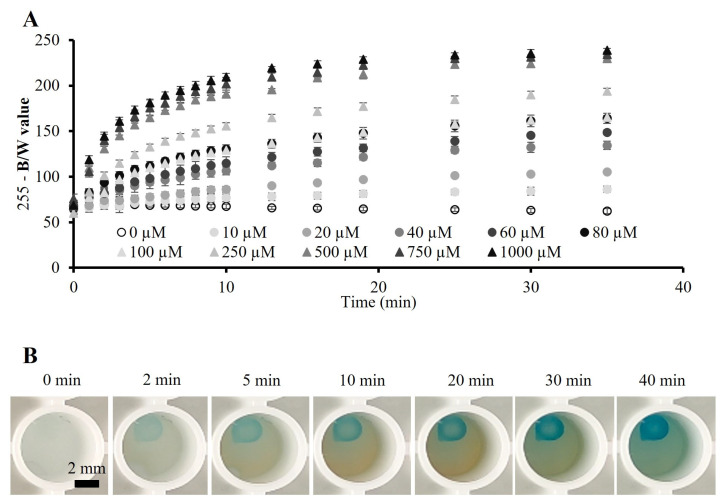
B/W analysis of the alginate beads for glucose sensing. (**A**) B/W values for glucose 10–1000 µM in the alginate biosystem for 40 min (n = 4). (**B**) Real images of an alginate bead biosystem in artificial sweat 80 µM glucose solution, over time.

**Figure 3 polymers-15-01218-f003:**
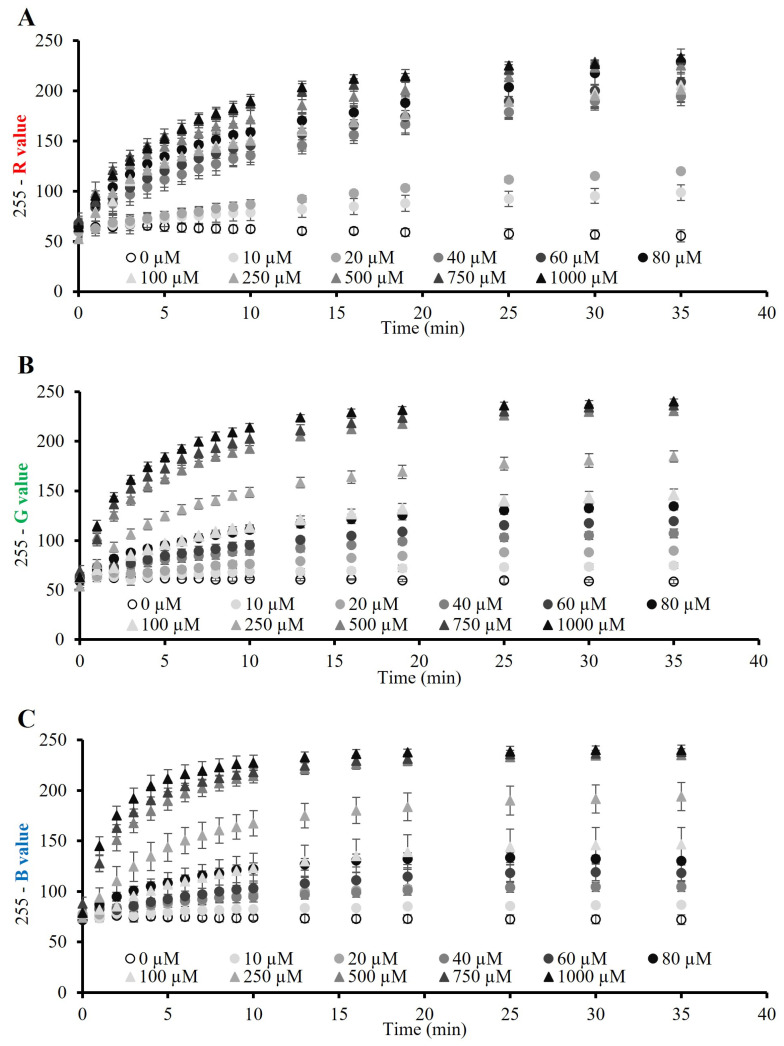
RGB analysis of the alginate beads for glucose sensing. (**A**) Red (R), (**B**) green (G) and (**C**) blue (B) values of the beads for glucose 10–1000 µM in the alginate biosystem for 40 min (n = 4).

**Figure 4 polymers-15-01218-f004:**
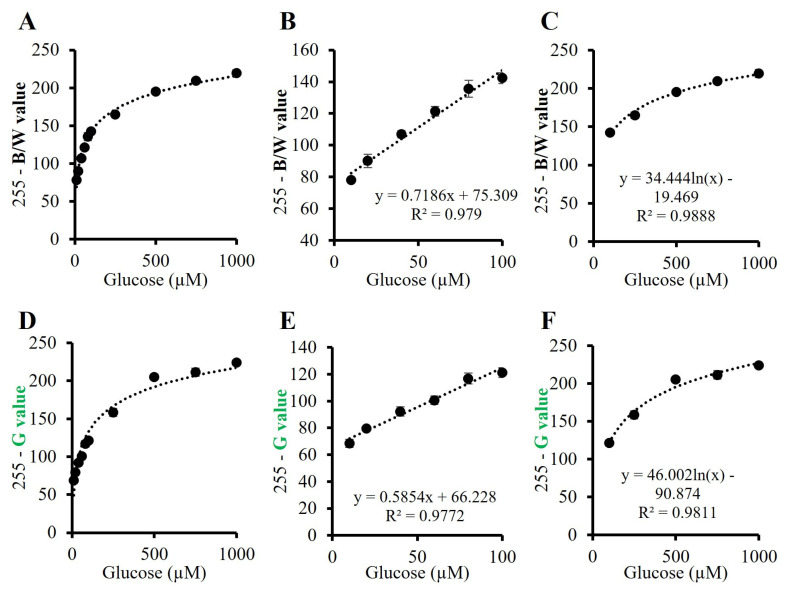
Calibration curves for glucose sensing in artificial sweat using the alginate biosystem. B/W values at 13 min of (**A**) glucose 10–1000 µM, (**B**) glucose 10–100 µM in a linear range and (**C**) glucose 100–1000 µM in a logarithmic fit. Green (G) values at 13 min of (**D**) glucose 10–1000 µM, (**E**) glucose 10–100 µM in a linear range and (**F**) glucose 100–1000 µM in a logarithmic fit (n = 4).

**Figure 5 polymers-15-01218-f005:**
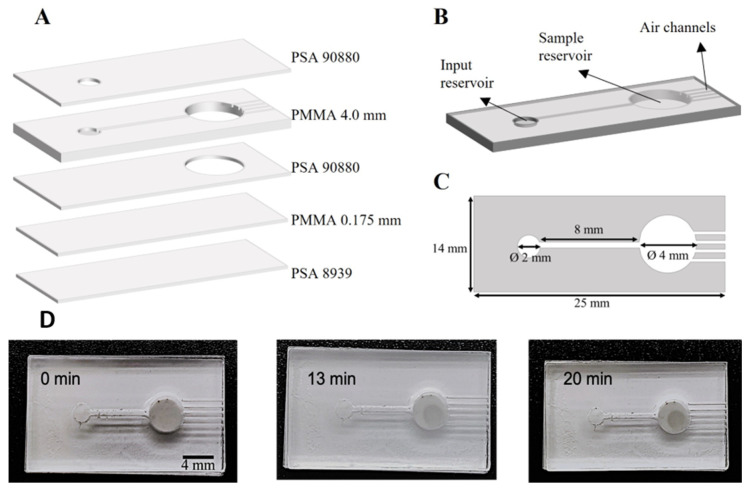
Microfluidic platform for the applicability of the biosystem with real sweat. (**A**) PMMA and PSA layers used for the fabrication of the microfluidic platform. (**B**) Schematic diagram of the microfluidic platform, which was 25 × 14 × 4.5 mm. (**C**) Dimensions of the microfluidic PMMA layer. (**D**) Pictures of the microfluidic platform with integrated biosystem to measure glucose in real sweat at 0 (left), 13 (center) and 20 min (right) (contrast was slightly increased in the pictures for a better appreciation of the beads). The intensity of the color increases over time due to a higher amount of glucose entering the bead.

## Data Availability

Not applicable.

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
