# Peer review of "Colorimetric Determination of Glucose in Sweat Using an Alginate-Based Biosystem"

_polymers, 2023, doi:10.3390/polym15051218_

Round 1

Reviewer 1 Report

In this manuscript, the author are presenting an alginate-based biosensor for the determination of glucose concentration in sweat, both in a plate format and in a microfluidic system.

The results presented here are of interest for the readership of Polymers, and fit well inside the special issue « medical applicationsof polymer-based composites ». The presented sensing strategy is rather simple and still powerful, looking at the relevant LOD claimed by the authors. However, several controls are missing which would strongly improve the results of this manuscript. I would therefore recommend a major revision, and I’m happy to review again this study after the completion of the following controls and the addition of the major points below:

-        -  The microfluidic format is a dynamic assay, with possible loss of enzymes in the stream. It’s not obvious that the calibration curves used in plates could be directly used for this microfluidic format. Therefore, it would be necessary to perform a calibration curve using artificial sweat in the microfluidic format in order to know if the two assays are equivalent.

-        - Then, determination of the glucose concentration in real sweat might not be possible using the calibration curves with artificial sweat, as inhibiters of the assay enzymes could be present in the more complex real sample. A quick experiment with spiked glucose in real sweat will help to answer this question. I’m afraid you must find some extra healthy cyclist to run some extra experiments…

-        - Finally, a control with another technique is still necessary when claiming results with real sweat. Would it be possible to concentrate the sweat sample to make it work with the glucometer ? For example, drying a sample of real sweat and resuspend it in a volume 1000x lower?

-         - Could you discuss the advantages (or withdrawals) of alginate beads for (long-term) storage of the enzymes ?

-         - l.173 : what is the pore size of the alginate network ? In particular, a discussion on the size difference between the pores and the enzymes would be useful, and it would be perfect to get an idea of the diffusion coefficient of the glucose and enzymes in this network.

-        - Page 6, the discussion on the color channels would be way more clear with the following information :

o   What are the absorption spectra of the two oxydation products ?

o   How is the camera working ? Is it 3 filters (centered on the green/blue/red wavelengths) in front of each individual color pixels ? What would be the width of this filters ? That would be necessary to understand why the green channel in ImageJ is better.

o   Finally, is the B/W intensity just the sum of the 3 color intensities ?

-          

Minor points :

-         - l. 106 : is the concentration of the sodium alginate as % w/w or % v/v ?

-         - l.111 : what do you mean as « wiped again » ?

-        -  the section 2.5 would be better placed at the line 116

-       -   l. 137 : what are the thicknesses of the PSA layers ?

-        - l. 147 : what material was used for the capillaries ?

-       -  l.153 : is this glucometer used for glucose analysis in blood ? Is it normally used for sweat analysis ?

-        - section 2.5 : some details are missing. In particular, are you only measuring the intensities of the beads or also the surrounding media ? For the beads, are the final values the mean value of all the pixels of the bead ?

-        - l.281 : could you discuss if there is any difference of sample volume with the cited technique, and if the LOD is for artificial or real sweat ?

-        - Figure 6 : I could not see any blue color on these pictures…

Author Response

REVIEWER 1

In this manuscript, the authors are presenting an alginate-based biosensor for the determination of glucose concentration in sweat, both in a plate format and in a microfluidic system.

The results presented here are of interest for the readership of Polymers,  and fit well inside the special issue « medical applications of polymer-based composites ». The presented sensing strategy is rather simple and still powerful, looking at the relevant LOD claimed by the authors. However, several controls are missing which would strongly improve the results of this manuscript. I would therefore recommend a major revision, and I’m happy to review again this study after the completion of the following controls and the addition of the major points below:

  • The microfluidic format is a dynamic assay, with possible loss of enzymes in the stream. It’s not obvious that the calibration curves used in plates could be directly used for this microfluidic format. Therefore, it would be necessary to perform a calibration curve using artificial sweat in the microfluidic format in order to know if the two assays are equivalent.

Thank for the comment. We agree with the reviewer that a calibration in device would be needed for a proper validation of the performance of the microfluidic platform with real sweat. However, the assay we performed with real sweat in device was not intended to be a validation of the biosystem, but rather a demonstration of the applicability if this alginate-based biosystem for real case scenarios. That is the reason why we decided to write the following heading for section 3.4 (line 308) “Integration and application of the biosystem in a microfluidic platform with real sweat”, rather than “integration and validation”. As the reviewer stated, the biosystem in plate and in device could show slightly differences regarding its performance. For future research, more exhaustive research would need to be done, including the calibration in device. This would improve the impact that the research would have in the sports field.

  • Then, determination of the glucose concentration in real sweat might not be possible using the calibration curves with artificial sweat, as inhibiters of the assay enzymes could be present in the more complex real sample. A quick experiment with spiked glucose in real sweat will help to answer this question. I’m afraid you must find some extra healthy cyclist to run some extra experiments…

Thanks for the comment. We agree that more exhaustive research should be done for a proper validation of the device with real sweat. However, as mentioned in the previous comment, we did not intend for a validation of the microfluidic platform with real sweat, but rather a demonstration of its application in a microfluidic platform. The microfluidic platform is not the focus of the manuscript but rather an example of the possible integration of the biosytem in a device format.

  • Finally, a control with another technique is still necessary when claiming results with real sweat. Would it be possible to concentrate the sweat sample to make it work with the glucometer? For example, drying a sample of real sweat and resuspend it in a volume 1000x lower?

Thanks for the comment. This is a really good appreciation since it would help to improve the detection of low concentrated samples and thus, the efficiency of the biosystem. In fact, we already tried concentrating the sample by drying a real sweat sample. However, after resuspending it, the volume needed to get a measurable concentration of glucose was of a few µL, which was not enough to perform the assays in device. This is one of the problems that arises when working with sweat in real case scenarios. Sweat is produced by the sweat glands in the range of nL - µL. Therefore, the amount of sweat that can be collected directly from the skin is limited by the inherent secretion activity of the glands, which complicates the sample collection in the microfluidic platforms attached onto the skin. We believe that the biosystem is applicable for sweat analysis but a redesign and further development of an appropriated microfluidic device will be need.

Considering the comments from the reviewer, we have added several comments in the manuscript to try to avoid the reader to have the sensation that the microfluidic section is the focus of the manuscript.

E.g. in the abstract: The biosystem was also applied with real sweat using a prototype of a microfluidic device platform, as a proof of concept. This research demonstrates the potential of alginate hydrogels as scaffolds for the fabrication of biosensors and their possible integration in microfluidic devices.

  • Could you discuss the advantages (or withdrawals) of alginate beads for (long-term) storage of the enzymes?

Thanks for the comment. Alginate hydrogels can, indeed, maintain the functionality of the enzymes, which we demonstrated in a previous work. We believe that this fact is a key feature that will allow alginate hydrogels to be considered for POC applications in sports science, although still more research is needed. We have included this information in the manuscript and modified the text as follows: (lines 82 – 85) “Moreover, we have also demonstrated the stability of an alginate-based biosystem integrated with an enzymatic reaction for the detection of glucose and lactate, in which the sensing capabilities of the material were maintained for, at least, 10 days [17]”.

The following reference was added to the text and the bibliography was changes accordingly: “[17]      Gunatilake, U.B., Garcia-Rey, S., Ojeda, E., Basabe-Desmonts, L., Benito-Lopez, F. TiO2 Nanotubes Alginate Hydrogel Scaffold for Rapid Sensing of Sweat Biomarkers: Lactate and Glucose. ACS Appl. Mater. Interfaces 2021, 13, 31, 37734–37745, https://doi.org/10.1021/acsami.1c11446”.

  • 173: what is the pore size of the alginate network ? In particular, a discussion on the size difference between the pores and the enzymes would be useful, and it would be perfect to get an idea of the diffusion coefficient of the glucose and enzymes in this network.

Thanks for the comment. The mesh size of the alginate biosystem was not measured in this research, but was previously reported to be between 6 – 14 nm. This information will help understanding the diffusion of the components of the enzymatic assay outside the alginate bead. Following the reviewer’s comment, we have modified the text as follows: (line 181 – 183) “This phenomenon can be better understand taking into consideration the mesh size of the alginate biosystem, which has been previously reported to be 6 – 14 nm [21]”.

The following reference was added to the text and the bibliography was changes accordingly: “[21]      Turco, G., Donati, I., Grassi, M., Marchioli, G., Lapasin, R., Paoletti, S. Mechanical Spectroscopy and Relaxometry on Alginate Hydrogels: A Comparative Analysis for Structural Characterization and Network Mesh Size Determination. Biomacromolecules 2011, 12, 4, 1272–1282, https://doi.org/10.1021/bm101556m”.

  • Page 6, the discussion on the color channels would be way more clear with the following information:
    • What are the absorption spectra of the two oxidation products?

This data is already mentioned in the text (section 3.1, line 186 – 192): “The first colored product, the resulting product of the one-electron oxidation, consists of a charge-transfer complex formed by the diamine, its oxidized diimine product and the radical cation, both species existing in equilibrium. This product absorbs light at 370 nm and 652 nm respectively, which provides the material with its characteristic blue color. Moreover, it can undergo a further two-electron oxidation, yielding a diamine, which absorbs visible light at 450 nm, generating an orange/yellow-colored product [22]”.

  • How is the camera working? Is it 3 filters (centered on the green/blue/red wavelengths) in front of each individual color pixels? What would be the width of this filters? That would be necessary to understand why the green channel in ImageJ is better.

Images of the beads were taken with a “64 MP camera (Sony IMX682 1/1.73”, f/1.89, PDAF) in a white chamber under the same light conditions”, as stated in Section 2.3. The B/W, R, G, and B analysis were performed individually using the same image, which was modified accordingly with ImageJ. For the B/W analysis, the image was turned into an 8-bit image, while the RGB analysis was done by dividing the original image into three independent images, one for each of the channels of the RGB color code. To provide the reader with a better understanding regarding the colorimetric analysis of the beads, we have included the following sentence in the manuscript: (line 124 – 125) “For each specific time, the same image was used for the four colorimetric analysis and was modified accordingly with ImageJ”.

  • Finally, is the B/W intensity just the sum of the 3 color intensities?

In this regard, RGB (0,0,0) refers to white, and RGB (255,255,255) refers to black. We have modified the manuscript as follows to provide this information: (line 130 – 131) “It needs to be noted that RGB (0,0,0) stands for white while RGB (255,255,255) stands for black”.

Minor points:

Thank you for your comments, which will allow the improvement of the manuscript. We have addressed these minor points one by one.

  • 106: is the concentration of the sodium alginate as % w/w or % v/v?

The concentration of the sodium alginate was % w/v in distilled water. To provide the reader with a better understanding of section 2.2 of the main manuscript, we have modified the text as follows: (line 108 – 109) “with 30 µL of sodium alginate (Sigma-Aldrich) 1.5 % (w/v) in distilled water”.

  • 111: what do you mean as « wiped again »?

This was a mistake we made during the preparation of the manuscript, and we just meant to say “wiped”. We have modified the text as follows: (line 113 - 114) “Afterwards, the newly formed beads washed with distilled water for 3 min before being wiped”.

  • the section 2.5 would be better placed at the line 116

We agree with the reviewer. We have placed this part of the manuscript inside section 2.3 of the manuscript (lines 120 – 132) and we have modified the text accordingly.

  • 137: what are the thicknesses of the PSA layers?

The total thickness of PSA 90880 and PSA 8939 is 244 µm and 229 µm, respectively. This information was added to the manuscript: (line 151 – 155) “The different layers of the device were, from bottom to top, 229 µm-thickness PSA 8939 (ARcare® 8939, Adhesive Research, Ireland), 0.175 mm-thickness PMMA (ME30-SH-000116, clear, Goodfellow, United Kingdom), 244 µm-thickness PSA 90880 (ARseal™ 90880, Adhesive Research, Ireland), 4.0 mm-thickness PMMA (ME303040, clear, Goodfellow, United Kingdom) and PSA 90880”.

  • 147: what material was used for the capillaries?

The capillaries were made of silicone. This information was added to the text as follows: (line 164) “through transparent silicone capillary tubing”.

  • 153: is this glucometer used for glucose analysis in blood? Is it normally used for sweat analysis?

The glucometer used in this research was, indeed, indicated for blood analysis. However, there are not commercially viable, non-invasive glucose monitors on the market at the moment [1]. In fact, the lack of commercial glucometers hinders the proper development of novel platforms for sweat sensing when comes to validated the results and compare them with existing technology. Nevertheless, comparison of sweat glucose levels with commercially available blood glucometers has been previously reported, like reference [24].

  • section 2.5: some details are missing. In particular, are you only measuring the intensities of the beads or also the surrounding media? For the beads, are the final values the mean value of all the pixels of the bead?

The analysis of the beads was carried out measuring the whole bead, without taking into consideration the surrounding media. This information was added to the text as follows: (line 131 – 132) “F or the B/W, R, G and B analysis, the whole bead was measured, without including the surrounding media”.

  • 281: could you discuss if there is any difference of sample volume with the cited technique, and if the LOD is for artificial or real sweat?

Both investigations mentioned in this section (references [23] and [24] of the manuscript), do not report the sample volume. They just mention that they place the device on the skin and proceed with the assay. Moreover, the LOD they mention have been calculated prior using the assays with human samples.

  • Figure 6: I could not see any blue color on these pictures…

The measured glucose concentration was very low, near the lowest glucose concentration used for the calibration of the biosystem with artificial sweat. Consequently, the blue intensity of the bead was also low and cannot be properly appreciated by naked eye. We have merged Figure 5 and 6 and modified the text as follows: (line 345-348: Figure 5D shows real images of the detection of glucose inside the device at 0, 13 and 20 min, in which the increasing in color intensity of the beads is appreciable as an increase of opacity of the bead. The truly blue color of the bead is not appreciable by the naked eye in this experiment, mainly due to the low concentration of glucose in the sweat sample.).

Reviewer 2 Report

As the authors refers further research should be done in order to proposed technology reach the potential end user. However, the results of this work shows that this is a very promisor system for the analytical determination of glucose.

 Authors should revise the text in order to correct some typos such as:

- Line 99: instead of “y” authors should write “and”

- Line 103: include a comma after (HRP)

Author Response

REVIEWER 2

As the authors refers further research should be done in order to proposed technology reach the potential end user. However, the results of this work shows that this is a very promisor system for the analytical determination of glucose.

Authors should revise the text in order to correct some typos such as:

  • Line 99: instead of “y” authors should write “and”
  • Line 103: include a comma after (HRP)

Thank you for your comment, we appreciate your evaluation of our research. We agree with the reviewer that more research is needed in order for this biosystem to be applied in real case scenarios when using microfluidic devices. However, we believe that the results showed in this manuscript will be of benefit for the development of future microfluidic platforms for sweat sensing. As the reviewer stated, we have modified the text accordingly as follows: (line 102) “750 and 1000 µM” and (line 106 – 107) “5 µL of glucose oxidase (GOx) 0.8 mg mL-1 (Aspergillus niger, Sigma-Aldrich, Spain), 5 µL of horseradish peroxidase (HRP) 0.04 mg mL-1 (Sigma Aldrich, Spain), and 1.5 µL of tetramethylbenzidine (TMB)”.

Reviewer 3 Report

Blood glucose detection is of great significance. Sweat-based blood glucose detection has attracted much attention due to its non-invasive characteristics. In this paper, a method based on colorimetric detection is established. Combining with microfluidic technology, it is expected to form a potential POCT device for blood glucose detection. However, there are still some points that need further clarification in this study.

1. A few text errors, for example, in the Line 68, "seat glucose" may be "sweet glucose".

2. Based on the fact that the detection effect of green is better than that of blue, the author believes that it is the result of two-step reaction. Here, in-depth analysis is needed, especially from the calculation and discussion of the change of substance concentration in chemical reaction.

3. The description of the detection based on microfluidic chip in section 3.4 is not clear enough. For example, it is necessary to explain the purpose of each component in the chip, the sample flow and analysis process on the chip. The optical detection results on the chip are analyzed using the non-chip calibration results. Whether the results of these two experimental detection methods are comparable needs to be tested and discussed. What is the result of comparison with commercial equipment?

Author Response

REVIEWER  3

Blood glucose detection is of great significance. Sweat-based blood glucose detection has attracted much attention due to its non-invasive characteristics. In this paper, a method based on colorimetric detection is established. Combining with microfluidic technology, it is expected to form a potential POCT device for blood glucose detection. However, there are still some points that need further clarification in this study.

Thank you for your comments, we agree with the reviewer regarding the potential of this biosystem as a POC platform, which could be applied both for blood and sweat glucose detection. However, extensive research needs to be done in order to develop an appropriate microfluidic platform with applications in the sports field, and it would need to consider sample collection avoiding evaporation and cross-contamination, sweat transportation inside the device and on-body performance, among others.

  1. A few text errors, for example, in the Line 68, "seat glucose" may be "sweet glucose".

We have modified the text as follows: (line 68) “…measuring sweat glucose in a linear…”

  1. Based on the fact that the detection effect of green is better than that of blue, the author believes that it is the result of two-step reaction. Here, in-depth analysis is needed, especially from the calculation and discussion of the change of substance concentration in chemical reaction.

We believe that this is a very interesting comment to enhance the understanding of the working principle of the biosystem, and we agree with the reviewer that more in-depth research would need to be done in this regard.  Since the color of the beads observed by naked eye was apparently blue, we believed that the measurement of the B values of the beads would yield the best analysis in RGB color code. However, after doing the analysis, we observed that it was the G analysis the one which yielded better results, rather than the B analysis, which led to big errors in the measured values and to a bad resolution of the different glucose concentrations. We conclude that a possible explanation for these results could be the two-step oxidation process that the TMB undergoes. However, we could not compare these results with previously published ones since we did not find in the consulted literature any research in which similar results were obtained. Therefore, we did not make that conclusion as a fully demonstrated statement, but rather as a possible explanation of this phenomenon that would need further research in order to be proven right.

  1. The description of the detection based on microfluidic chip in section 3.4 is not clear enough. For example, it is necessary to explain the purpose of each component in the chip, the sample flow and analysis process on the chip. The optical detection results on the chip are analyzed using the non-chip calibration results. Whether the results of these two experimental detection methods are comparable needs to be tested and discussed. What is the result of comparison with commercial equipment?

Thank you for this comment, we agree with the reviewer that the microfluidic device used in this research is not properly explained. Therefore, to enhance the understanding of the performance of the device and following the reviewer’s comment, we modified Figure 5 (page 11) and the text in section 3.4 as follows: (line 318 – 325) “The bottom PSA layer, which provided the device with a white color, was added to give a better contrast and to enhance the colorimetric analysis. The device was 25 x 14 x 4.5 mm and consisted of a 2 mm-diameter input reservoir, a 400 µm microfluidic channel, a 4 mm-diameter sample reservoir and five 100 µm air channels (Figure 5B). These were added to the design in order to avoid the formation of air bubbles inside the microfluidic platform, which allowed the air to flow out of the device while sweat flowed towards the sample reservoir. Figure 5C shows the dimensions of the microfluidic PMMA layer.”

As for the optical detection, we use the calibration in plate for the analysis of the on-chip results, as the reviewer stated. We did not carried out the calibration on-chip since we did not intend to validate this biosystem in real case scenarios, for which a more exhaustive research would be needed. Therefore, the operation of the device with real sweat was just carried out to demonstrate the applicability of the alginate-based biosystem developed in this research with real samples. That is the reason why we decided to give this section the following heading (line 308) “3.4 Integration and application of the biosystem in a microfluidic platform with real sweat”, rather than “integration and validation”. We tried to compare the measured value with a commercial glucometer. However, it was out of range, and we could not make a clear conclusion in this regard, which we have explained in the text as follows: (line 351 – 356) “The measured concentration was compared with a commercially available glucometer (see Section 2.4). The glucometer range was 1100 – 27800 µM, which was much higher than the tested glucose concentrations of the real sweat sample thus it did not yield to any valid result. However, the fact that it was able to obtain a measurable result with the alginate biosystem, using real sweat, demonstrated the applicability of this biosystem as a proof-of-concept for the determination of glucose in real field scenarios”.

References

[1] Zafar, H.; Channa, A.; Jeoti, V.; Stojanović, G.M. Comprehensive Review on Wearable Sweat-Glucose Sensors for Continuous Glucose Monitoring. Sensors 2022, 22, 638. https://doi.org/10.3390/s22020638.

Round 2

Reviewer 3 Report

The manuscript can be published in the present form.